# 3D architecture and complex behavior along the simple central San Andreas fault

Yifang Cheng [1,2,3,4] ✉, Roland Bürgmann [2,3] & Richard M. Allen [2,3]

The central San Andreas Fault (CSAF) exhibits a simple linear large-scale fault geometry, yet seismic and aseismic deformation features vary in a complex way along the fault. Here we investigate fault zone behaviors using geodetic observation, seismicity and microearthquake focal mechanisms. We employ an improved focal-mechanism characterization method using relative earthquake radiation patterns on 75,164 $M_l \geq 1$ earthquakes along a 2-km-wide, 190-km-long segment of the CSAF, from 1984 to 2015. The data reveal the 3D fine-scale structure and interseismic kinematics of the CSAF. Our findings indicate that the first-order spatial variations in interseismic fault creep rate, creep direction, and the fault zone stress field can be explained by a simple fault coupling model. The inferred 3D mechanical properties of a mechanically weak and poorly coupled fault zone provide a unified understanding of the complex fine-scale kinematics, indicating distributed slip deficits facilitating small-to-moderate earthquakes, localized stress heterogeneities, and complex multi-scale ruptures along the fault. Through this detailed mapping, we aim to relate the fine-scale fault architecture to potential future faulting behavior along the CSAF.

Fault zones have geometric complexities and are multiscale systems consisting of a localized fault core accommodating the primary slip (on-fault), a highly fractured damage zone around the fault core accommodating a small fraction of deformation (off-fault), and the surrounding host rock[1–3]. Fault zones can deform seismically, such as in large damaging earthquakes, small earthquakes including earthquake clusters and repeating events, and tectonic tremor[4], and aseismically, such as by transient slow slip events, steady creep, and afterslip[5,6]. The partitioning of seismic and aseismic fault slip is usually quantified using geodetically determined fault coupling, which is defined as the ratio of the inferred slip deficit rate and the long-term slip rate, with a value of 1 corresponding to fully locked while a value of 0 indicates a freely creeping fault. The spatial distribution of fault coupling and implied slip deficits can help us better understand fault zone properties, determine the seismic potential of faults, and constrain the return period and the maximum-possible magnitude of earthquakes[7–9]. Fault sections with larger kinematic coupling at depth generally correspond

to lower surface creep rates observed from geodetic investigations[10–13], lower recurrence rates of repeating earthquakes[14–16], lower b-values[17], and a larger fraction of clustered events[18].

However, gaps remain in our understanding of the seismic and aseismic deformation of faults due to several problems. It is often assumed that all patches on a fault slip in the same direction without considering the variation of slip directions that we can expect in areas with abrupt changes of fault geometry or creep rate[19]. Secondly, analyses of seismicity often focus on statistical parameters estimated from earthquake occurrences instead of the underlying physical processes. Moreover, the effects of distributed fault-zone deformation and on-fault/off-fault interactions are usually ignored, although on-fault and off-fault deformations are tightly interlinked and coevolve with strong feedback loops over multiple spatial and temporal scales[20] (Fig. 1a). These challenges can be potentially overcome by analyzing focal mechanisms of small earthquakes, which helps to resolve fine-scale on-fault slip directions, provides physical parameters for seismicity

[1]State Key Laboratory of Marine Geology, Tongji University, Shanghai, China. [2]Department of Earth and Planetary Science, University of California, Berkeley, CA, USA. [3]Berkeley Seismological Laboratory, University of California, Berkeley, CA, USA. [4]School of Ocean and Earth Science, Tongji University, Shanghai, China. ✉e-mail: chengyif@tongji.edu.cn

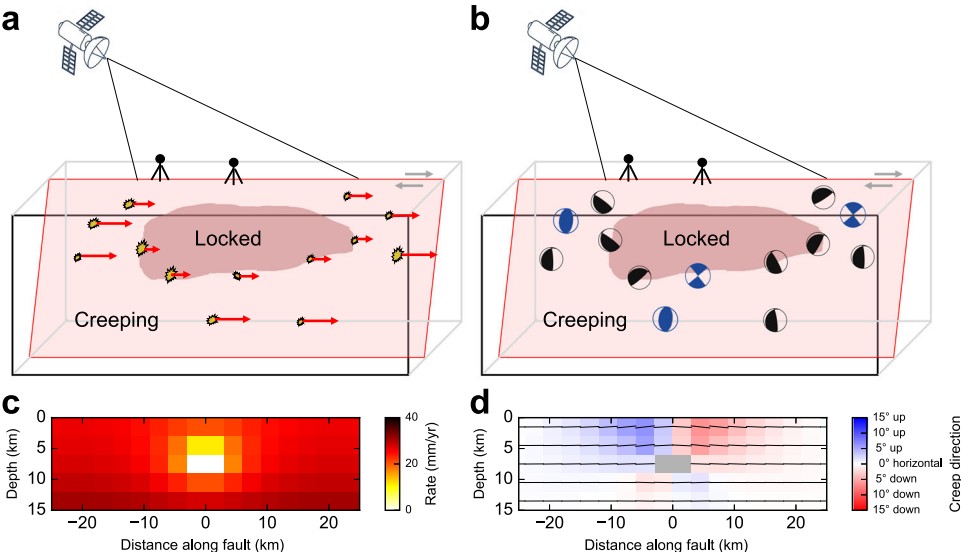

**Fig. 1 | Improvements in constraining subsurface coupling using earthquake focal mechanisms.** Schematic comparison of data and assumptions used in **a** previous geodetic inversion studies[11–13,15] and **b** this study relying on the spatial distribution of earthquakes and slip directions from focal mechanisms. Dark red areas denote locked patches and light red areas denote creeping areas. Modeled fault **c** creep rate and **d** creep direction around a locked patch. Yellow stars and beachballs denote earthquake locations and focal mechanisms, respectively. Black and blue beachballs are on-fault and off-fault earthquakes, respectively. Red arrows show the assumed fault slip direction. Black thin lines in (**d**) denote local creep directions on the NE side of the fault.

analysis (e.g., fault and stress orientations), and allows for differentiating on- and off-fault seismic deformation (Fig. 1b). Furthermore, if a locked patch exists at depth, the elastic fault slip modeling results (see Methods section) show that the creep rate slightly decreases (Fig. 1c) and the creep direction rotates around the locked patch (Fig. 1d), suggesting that focal mechanisms can provide crucial information on the finer details of fault coupling that cannot be resolved from geodetically measured surface displacements. In this work, we integrate a comprehensive catalog of focal mechanisms to deepen our understanding of the fine-scale architecture and complex behaviors of major fault zones.

The 190-km-long central segment of the San Andreas Fault (CSAF) in California (Fig. 2) offers a unique opportunity to investigate the fine-scale complexities of seismic and aseismic fault zone processes and their interconnected nature. The CSAF is characterized by a long-term slip rate of ~33–35 mm/yr and varying surface creep at ≤30 mm/yr[21,22]. It lacks large historical earthquakes, while $M_w$ ~7–8 earthquakes repeatedly occur to the north and south (e.g., the 1906 San Francisco earthquake and the 1857 Fort Tejon earthquake; Fig. 2a and Supplementary Fig. 1). Understanding the potential for large earthquakes along the CSAF is important for seismic hazard assessment of the SAF and other large creeping fault zones in the world[23]. The CSAF also provides a natural laboratory for studies of seismic and aseismic fault zone processes because of the availability of a dense, continuous monitoring network. Decades of monitoring show that the CSAF exhibits heterogeneous seismic and aseismic deformation patterns and variable fault coupling inferred from varying along-fault surface displacements[11–13,21], the occurrence of moderate earthquakes (e.g. 1922, 1934, 1966, and 2004 $M_w$6.0 Parkfield earthquake), abundant small earthquakes[24], and repeating earthquakes whose recurrence rate is proportional to the fault creep rate at depth[25–28]. In order to obtain focal mechanisms of the above-mentioned small earthquakes, we use a recently developed relative focal mechanism calculation algorithm[29] and obtained high-quality focal mechanisms (uncertainty <35 degree) of ~80% of the $M_l ≥ 1$ catalog events[24].

In this study, we integrate seismically measured subsurface fault zone processes, geodetically measured surface displacements, and modeling of strike-slip creeping fault to comprehensively characterize

and understand multiscale fault zone deformation processes. We apply our approach to the CSAF in California. Our results show that all observed fine-scale kinematic features of the fault zone can be reconciled with a mechanically weak and poorly coupled fault zone.

## Results
### Overview of seismicity data
In this study, we integrate the location, time, magnitude, and focal mechanism of 75,164 $M_l ≥ 1.0$ earthquakes, 145 $M_l ≥ 4.0$ earthquakes, and 355 repeating earthquakes from 1984 to 2015 to place high-resolution constraints on the fault structure of the CSAF (Fig. 3). Most $M_l ≥ 4.0$ sequences are located along the San Juan Bautista (SJB) and Parkfield (PK) transition zones and a few $M_l ≥ 4.0$ earthquakes are located near Bitterwater (BW). In contrast, repeating earthquakes occur more frequently between Melendy Ranch (MR) and the San Andreas Fault Observatory at Depth (SAFOD), implying high aseismic creep rates. Since most repeating earthquakes are concentrated in a narrow zone and appear to delineate the major fault strands, we first use repeating earthquakes to obtain the horizontal location of major fault strands and then use both repeating earthquakes and the surrounding $M_l ≥ 1.0$ earthquakes within 1 km epicentral distance from the horizontal location of major fault strands (Fig. 3a, b) to estimate the location and the orientation of the 3D main fault surface using principal component analysis[30] (see Methods section). The whole fault is nearly vertical with strike angles varying between N120°E to N140°E (Fig. 3d). Our final catalog includes 99.7% of the repeating earthquake sequences and 97.8% of $M_l ≥ 4.0$ earthquakes, as well as 83.4% of $M_l ≥ 1.0$ events located within 1 km NE from the main fault strand (Fig. 3e and Supplementary Fig. 2). To better quantify the fault zone structure, we select all focal mechanism solutions with uncertainty less than 25 degrees and pick the focal mechanism nodal plane closer to the main fault orientation and calculate the azimuthal difference between the nodal plane and the main fault. Note that this calculation assumes that the nodal plane closer to the main fault orientation is the real fault plane and ignores left-lateral faults at high angles to the main fault. For these near-fault earthquakes, we obtain the percentages of $M_l ≥ 4.0$ earthquakes, repeating earthquake sequences, and all $M_l ≥ 1.0$ earthquakes (including $M_l ≥ 4.0$ earthquakes and repeating

**a**

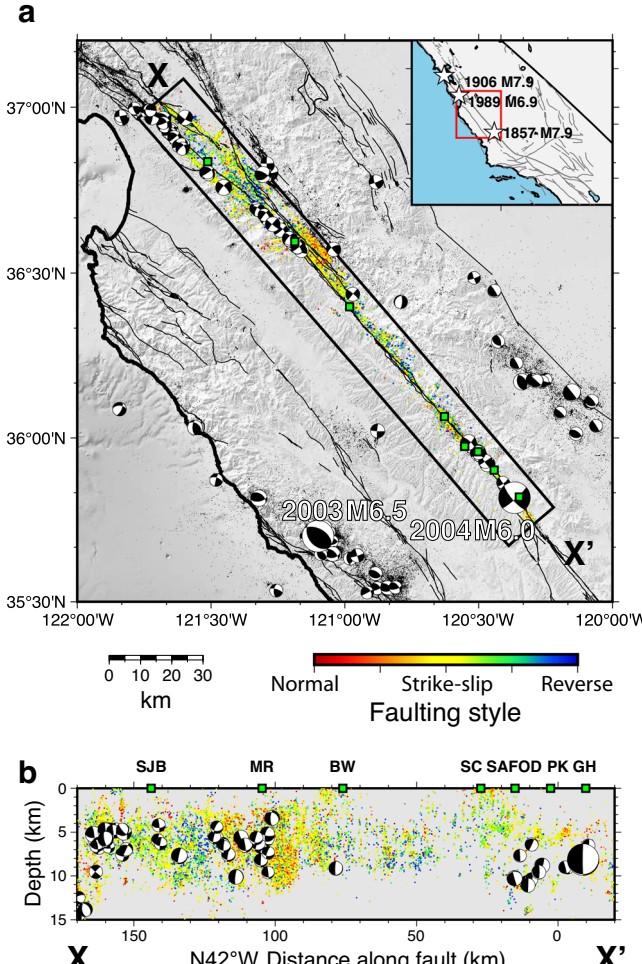

**Fig. 2 | Seismicity distribution along the CSAF. a** Map view and **b** cross-section view of relocated earthquakes from 1984 to 2015 (Waldhauser and Schaff[24], extended to later years) along the central San Andreas Fault. Events in the box XX' are colored by faulting style. Small beach balls denote $M_l \geq 4.0$ earthquakes. Large beach balls denote the 2003 $M_w$6.5 San Simeon and 2004 $M_w$6.0 Parkfield earthquakes' focal mechanisms. White stars denote historic major earthquakes. Note vertical exaggeration VE = 3.13 in (**b**). The local coordinate system has its origin at latitude 35.867°N, longitude 120.447°W and is oriented N42°W. SJB San Juan Bautista; MR Melendy Ranch; BW Bitterwater; SC Slack Canyon; SAFOD San Andreas Fault Observatory at Depth; MM Middle Mountain; PK Parkfield; GH Gold Hill.

earthquakes) with less than 20° azimuthal difference from the main fault, which are 92.9%, 94.1%, and 83.3%, respectively (Fig. 3f). The diversity of earthquake focal mechanisms increases gradually with decreasing magnitude even between $M_l$3.0 to 5.0 (Supplementary Fig. 3), suggesting that this trend is not caused by the uncertainty of focal mechanism solutions. Therefore, in the following analysis, we assume that $M_l \geq 4.0$ earthquakes and repeating earthquake sequences are located on the main fault (see Methods section; Supplementary Fig. 4), while $M_l \geq 1.0$ earthquakes have variable orientations and are located both on- and off-fault. To focus on the interseismic period and disregard the co- and early postseismic deformation due to the 1989 M6.9 Loma Prieta and 2004 $M_w$6.0 Parkfield earthquakes, we exclude earthquakes within the first 3 years after these events when calculating the occurrence rates of repeating earthquakes and focal mechanism properties.

## Fault model setup
The abundant small earthquake focal mechanisms provide a great opportunity to determine the fine-scale fault slip directions, which can

be directly compared with the output of kinematic fault models of the partially coupled CSAF employing shear-stress-free boundary conditions on the fault. As shown in Figs. 1c, d, if there is a locked fault patch at depth, the fault slip rate decreases slightly around the locked patch, and the surrounding slip directions exhibit opposite rotations near the two ends of the locked patch, providing a powerful constraint on the location and size of the locked patches at depth. Therefore, in this study, we forward model the interseismic fault deformation of the CSAF and compare the obtained slip kinematics with the fault zone properties estimated from seismicity.

Based on the definition of fault coupling and previous observations, most moderate-to-large earthquakes are located in high-coupling areas while most repeating earthquakes are interpreted as small patches that repeatedly break and are surrounded and driven to failure by aseismically slipping sections of the fault (low-coupling areas). Therefore, based on the spatial distribution of $M_l \geq 4.0$ events and repeating earthquakes (Fig. 3a, b), we design a simple forward model that consists of several locked patches near SJB, BW, and PK (in areas of low earthquake-density and near the $M_l \geq 4.0$ earthquakes; Fig. 3b) and an otherwise freely slipping shallow fault in the top 15 km depth driven by a buried fault plane with a 34 mm/yr interseismic deep creep rate[22] beneath it (model A in Fig. 4a). The fault is bounded by fully locked fault segments at the two ends, corresponding to the locked sections of the SAF that hosted the 1857 M7.9 Fort Tejon to the SE as well as the 1906 M7.8 San Francisco and 1989 M6.9 Loma Prieta earthquakes to the NW. We then compute the fault displacement rate and slip directions on the freely slipping patches using a boundary element method (Poly3D; see Methods section)[31].

## Observed and modeled along-fault displacements
We compare the forward modeling results with multiple geophysical observations. Besides the traditionally used surface creep rate (red curve and green symbols in Fig. 4f)[11,21,32], we also estimate the 2D variations of creep rate and creep direction on the NE side of the main fault using the occurrence rate and the slip direction of repeating earthquakes, respectively (see Methods section; Supplementary Figs. 4–6). Figure 4 shows the comparison of the modeled and observed creep rate using the occurrence rate of repeating earthquakes (Fig. 4b, c), the modeled and observed creep direction using the focal mechanisms of repeating earthquakes (Fig. 4d, e), and the modeled and observed surface creep rate using creepmeter, alignment array and InSAR data (Fig. 4f)[11,21,32]. The modeled results are overall consistent with the observed subsurface fault creep rate variations, the fault slip directions at depth, and the surface creep rates with correlation coefficients of 0.47 (Fig. 4b, c), 0.55 (Fig. 4d, e), and 0.91 (Fig. 4f), respectively (Figs. 6a–c). The presence of locked fault patches results in gradually decreasing creep rates around the patches. For example, the observed creep rates from repeating earthquakes and measured surface creep rates are lower than 25 mm/yr and 15 mm/yr, respectively, near the SJB and PK transition zones (Figs. 4b, c, f). In the central creeping section from MR to the SAFOD, the modeled and observed creep rates approach the deep creep rate (~34 mm/yr). The repeater-derived creep directions show a more heterogeneous distribution around the locked patches with an upward creep to the NW of the patch and downward creep to the SE of the patch (Fig. 4d, e), providing valuable additional constraints on fault coupling at depth. For example, there is no clear evidence of the existence of locked fault patches near BW based on just the variation of creep rates (Fig. 4f). In contrast, the abrupt changes of the vertical creep component (Fig. 4d, e) and the observed $M_l \geq 4.0$ sequences near BW (Fig. 4a) indicate considerable stress accumulation and the existence of a deep locked fault patch near BW.

## Off-fault structure and kinematics
In addition to on-fault displacements, there are also many small earthquakes located around the fault and most of them are located to

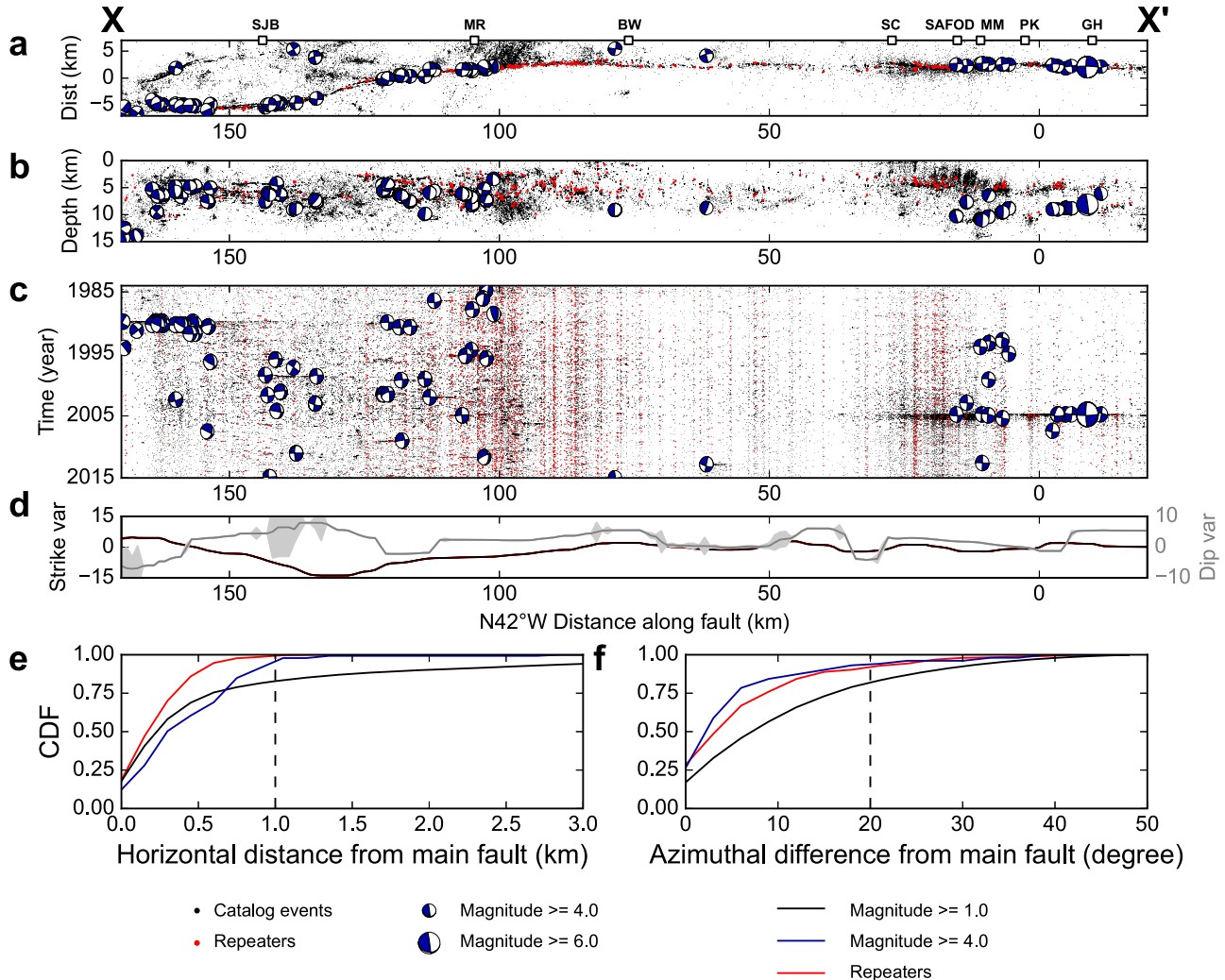

**Fig. 3 | Spatiotemporal variation of seismicity and its relationship with the main fault. a** Rotated map view, **b** cross-section view, and **c** spatiotemporal variations of seismicity (black dots), $M_l \geq 4.0$ earthquakes (blue beachballs), and repeating earthquakes (ref. 28; red dots) along the Central San Andreas Fault. The local coordinate system has its origin at latitude 35.867°N, longitude 120.447°W and is oriented N42°W. **d** Spatial variations of fault strike (red line) and dip (blue line) angles determined using PCA analysis from relocated seismicity around each 15 km-long 15 km-deep fault segment stepping at 1-km intervals along the fault. The reference fault strike and dip are N138°E and 90° (vertical), respectively. Cumulative Density Functions (CDF) of the **e** horizontal distance of events from main fault and **f** azimuthal difference from main fault strike for earthquakes located within 1 km horizontal distance from the horizontal location of main fault based on focal-mechanism nodal plane closest to the main fault. Black, blue, and red curves denote $M_l \geq 1.0$ earthquakes, $M_l \geq 4.0$ earthquakes, and repeating earthquake sequences, respectively.

the NE of the fault trace (Fig. 3 and Supplementary Fig. 2). Therefore, we obtain the along-fault cross-sectional distribution of the modeled azimuthal differences ($\theta$) between the off-fault maximum horizontal stress orientation ($\mathbf{SH_{max}}$) and the main fault 1.5 km NE of the main fault plane (see Methods section; Fig. 5a). We also observe a moderate correlation (0.33) between the modeled off-fault stress distribution and the observed stress field estimated using $M_l \geq 1.0$ focal mechanisms located within 2 km of the main fault (Figs. 5b and 6d), which is noteworthy considering the additional effects of background tectonic stress and the strong stress variations across the fault. Overall, $\mathbf{SH_{max}}$ shows low angle in the NW part of the fault (small $\theta$ value) but exhibits high angle in the SE part (large $\theta$ value). However, both modeled and observed variations of stress orientation near BW exhibit an opposite pattern with high angle between $\mathbf{SH_{max}}$ and fault strike (large $\theta$ value) to the NW of BW and low angle ($\theta$ value) to the SE of BW, consistent with the existence of a large deep locked patch near the BW. The consistency between the observed stress field and the modeled off-fault stress field variations suggests that on-fault coupling

heterogeneity can cause significant stress perturbation in the surrounding area.

We further investigate the off-fault structure and kinematics by calculating the cross-section distribution of the percentage of oblique-reverse faulting events (%$_{rake>0}$) (Fig. 5c). There is a significant negative correlation between $\theta$ and %$_{rake>0}$ with a correlation coefficient of $-0.43$ When $\mathbf{SH_{max}}$ is oriented at a high angle to the main fault, most events are oblique-normal faulting events with optimal fault orientations at a high angle to the main fault (case 1 in Fig. 5d, e). When $\mathbf{SH_{max}}$ is at about 45 degrees, there are a comparable number of oblique-normal and oblique-reverse faulting events, suggesting strike-slip faulting is the preferred faulting style (case 2 in Fig. 5d, e). When $\mathbf{SH_{max}}$ in the fault zone is at a low angle to the main fault, most earthquakes are oblique-reverse faulting events with optimal fault orientations showing high angles to the main fault (case 3 in Fig. 5d, e). In all cases, small-scale faults tend to have horizontal slip directions parallel to the main fault orientation and some of their fault strike orientations exhibit a high angle to the main-fault orientation.

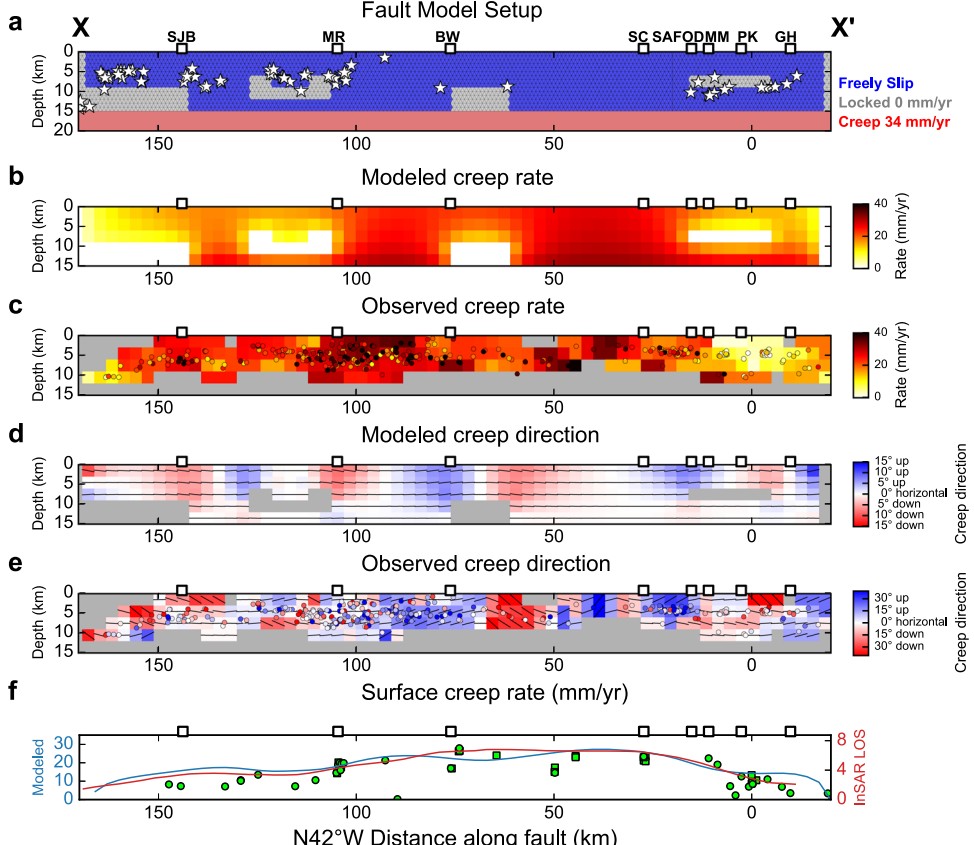

**Fig. 4 | Modeled and observed on-fault displacement. a** Fault model 1 setup with freely slipping zones (blue), locked sections (gray) and constant-rate creeping zones (red). The deep creeping zone driving the shallow creep extends from 15 km to 2000 km depth and far beyond the lateral ends of the CSAF. The shallow fault is fully coupled beyond 166 km NW and −17 km SE of the fault. The size of each fault patch is 3 × 3 km. White stars denote $M_I \geq 4.0$ earthquakes. **b** Modeled and **c** observed fault creep rate estimated from the occurrence of repeating earthquakes. **d** Modeled and **e** observed fault slip direction estimated from the rake and

dip of repeating earthquake focal mechanisms. Black thin line denotes local creep direction on the NE side of the fault. Colored dots and grids in (**c**, **e**) represent the values from each individual repeating earthquake sequence and those averaged in each spatial bin. Positive creep directions indicate a NE-side-up dip-slip component. **f** Modeled (blue) and observed surface creep rate (red) estimated from InSAR LOS data (Jolivet et al., 2014). Green circles and squares denote average surface creep rates from creepmeters and alignment arrays (circle: [32]; square: [21]), respectively.

## Discussion

To evaluate the proposed fault model, we also perform modeling using a simple creeping fault model without the small, locked patches but with fully locked segments on the two ends of the CSAF (model B, Supplementary Fig. 5a) and compared the modeling results using model B with the observations (Supplementary Fig. 6). Without the locked patches on the CSAF, the modeled creep rate is generally higher than the observed creep rate but the modeled and observed creep rates still show a first-order correlation (Supplementary Fig. 6a, c). In contrast, there is almost no correlation between the observed and modeled creep directions and stress orientations (Supplementary Fig. 6b, d), suggesting that the vertical fault slip component is highly sensitive to the small-scale fault coupling heterogeneity and can help to constrain the fault coupling model at depth. The solved fine-scale fault zone properties and fault coupling model provide great opportunity to improve our understanding of the physical mechanisms of small, locked asperities that were not noticed before.

There are some misfits between the modeled and observed fault zone properties due to the uncertainty of input data and the simplicity of the first-order fault coupling model. The range of modeled creep directions is ±6 degrees from the horizontal direction and is substantially smaller than that of the creep directions observed from repeating earthquake focal mechanisms (±30 degrees) (Fig. 6b). To explore the possible reasons for the observed misfits, we have performed a series of additional synthetic tests. The results show that

locked patches can cause stronger creep direction rotation when the patch is shallower (Supplementary Fig. 7) or larger (Supplementary Figs. 8 and 9), and the adjacent freely creeping area is larger (Supplementary Fig. 10). Many other factors that can intensify creep direction rotation are not captured in our model, such as deviations of fault segments from the vertical orientation (Supplementary Fig. 11), and the more substantial rotations near the locked patch's edge using models with finer grid sizes (Supplementary Fig. 12). The inferred rake angle on the fault plane can also change substantially due to local geometry variations and focal mechanism uncertainties (Hardebeck and Shearer, 2002; ~20 degree in this study), which can further affect the inferred creep direction. In contrast to the fault creep direction, the range of the modeled angle between $SH_{max}$ and fault strike direction ($\theta$) is 5–85 degrees is larger than the observed range (20–70 degrees) (Fig. 6d). This is because the inverted stress field is obtained from focal mechanisms of a large number of earthquakes ( >100) along a given fault patch including many on-fault earthquake focal mechanisms, compared to only a few (<10) repeating earthquake sequences in each fault patch, which is averaged over a long period and is less sensitive to focal mechanism uncertainties and local fault zone heterogeneities. Moreover, the observed surface creep rate has shown to be different from the modeled surface creep rate between MR and BW, which might be due to the temporal oscillation of surface creep rate or the simple first-order fault coupling model.

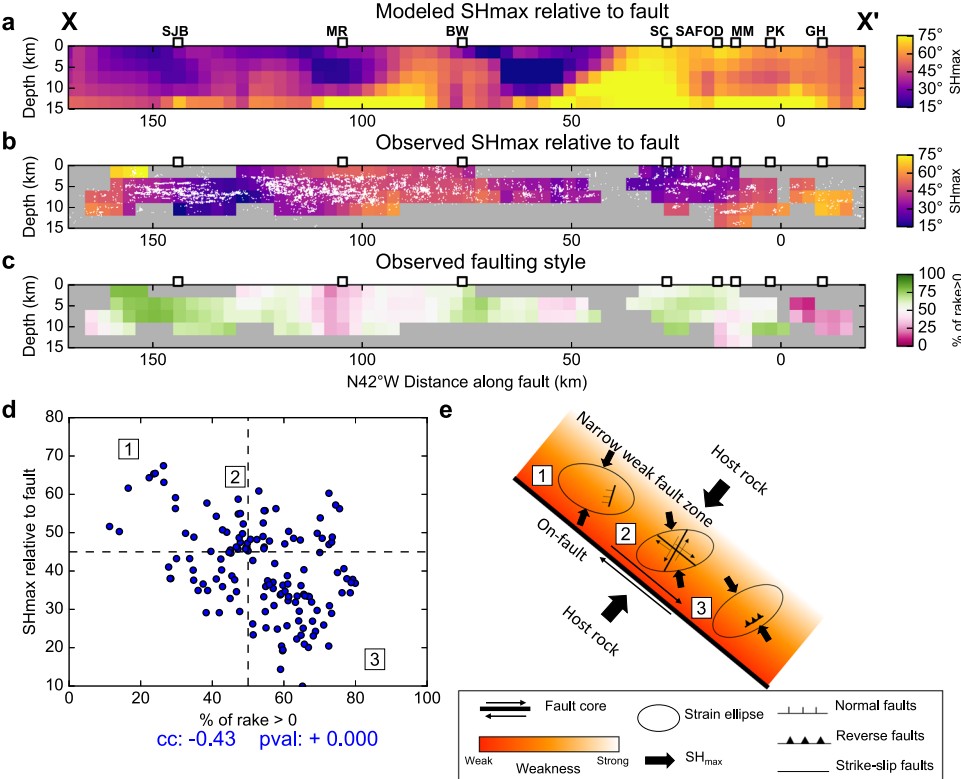

**Fig. 5 | Off-fault stress field and kinematics.** Comparison of **a** the angle between the main fault and the modeled off-fault maximum horizontal stress orientation ($\theta$) 1.5 km NE of the main fault with **b** the angle between the main fault and the observed maximum horizontal stress orientation calculated from $M_l \geq 1.0$ focal mechanisms located within 2 km around the fault trace. White dots denote the locations of focal mechanisms used in stress inversion. **c** The percentage of oblique-reverse-faulting events ($\%_{rake>0}$; red curve) for $M_l \geq 1.0$ focal mechanisms located within 2 km around the fault trace. **d** Point-to-point comparison of $\theta$ in (**b**) and $\%_{rake>0}$ in (**c**). **e** Schematic illustration showing the variation of fault zone structure, weakness, and stress field.

The creep rate distribution obtained from our first-order fault coupling model is overall consistent with those inverted from geodetic observations (Supplementary Fig. 14)[12,13], with low fault coupling on the central creeping section between 25 and 100 km along the fault and relatively high fault coupling on the other part of the fault, but with better-constrained fault slip distributions at depth. One interesting feature is that many of these models show a deep high-coupling area in the central creeping section but with somewhat different locations. In this study, we constrain the location of the deep locked patch near BW using additional observations from earthquakes at depth, including $M_l \geq 4.0$ earthquakes (Figs. 3 and 4a), reduced estimated creep rates (Fig. 4c), as well as abrupt changes of creep direction (Fig. 4e), off-fault stress orientation (Fig. 5b), and earthquake faulting style (Figs. 2 and 4c). Based on the modeling result, the differential creep rate between the locked patch near BW and the surrounding high-creep-rate patches is much higher compared with the other locked patches in the SJB and PK transition zones, suggesting a higher rate of elastic strain accumulation. However, there are only 2 instrumentally observed $M_l \geq 4.0$ ($M_w4.1$ and $M_w5.3$) earthquakes near the BW locked patch in the past 100 years compared with other locked patches near MR and PK. In contrast, there were six $M_w > 5.5$ earthquakes near BW between the 1857 Fort Tejon and the 1906 San Francisco earthquakes (Supplementary Fig. 1)[33]. The infrequent occurrences of large-magnitude earthquakes near the BW locked patch in the past century might be caused by the changes of absolute stress level and the recurrence times of moderate-to-large magnitude events caused by the occurrence of the 1857 Fort Tejon and 1906 San Francisco earthquakes[34]. Similar changes of the occurrence rate of large-magnitude earthquakes before and after the 1857 Fort Tejon and 1906 San Francisco

earthquakes are also seen along other sections of the fault (Supplementary Fig. 1)[33]. Since the main fault orientation is simple near the BW (Fig. 3a, d), the significant variations of fault coupling may be mainly due to the variations of material properties, such as lithology[35–37], local temperature anomalies[38,39], and pore fluid pressure[40]. In contrast, the local dip angle estimated from repeating earthquakes near SJB and PK exhibit about 20–30 degrees deviation from the major fault orientation, suggest that local fault geometry variation might considerably affect the fault coupling near the SJB and PK (Supplementary Fig. 15).

The seismic potential inferred from the fault coupling model along the CSAF is of interest from a hazard perspective because the CSAF accommodates nearly all the plate motion in this part of California. Assuming a deep slip rate of 34 mm/yr[21], the modeled moment deficit rate is $1.32 \times 10^{18}$ N·m/yr, while $1.48 \times 10^{18}$ N·m/yr are released by aseismic slip. The stored moment over a 150-year period is equivalent to a moment magnitude ($M_w$) 7.5 earthquake which agrees with previous studies[12,13]. If we assume that the areas of partial coupling catch up the moment deficit only by aseismic slip, such as by accelerated afterslip following earthquakes on the SAF, the modeled moment deficit rate is $4.04 \times 10^{17}$ N·m/yr and the 150-year accumulated moment is equivalent to a $M_w7.2$ earthquake. Since the locked patches are distributed along the creeping section with low slip deficit around them, they are more likely to rupture independently as moderate earthquakes (e.g., the foreshocks of the 1857 Fort Tejon earthquake and repeated occurrences of M6 Parkfield earthquakes; Supplementary Fig. 1) with a substantial amount of slip deficit on the surrounding fault being taken up by transient afterslip of major ruptures (e.g., following the 1906 Great San Francisco, 1989 Loma Prieta, and 2004 Parkfield earthquakes[15,27]). An important but

 

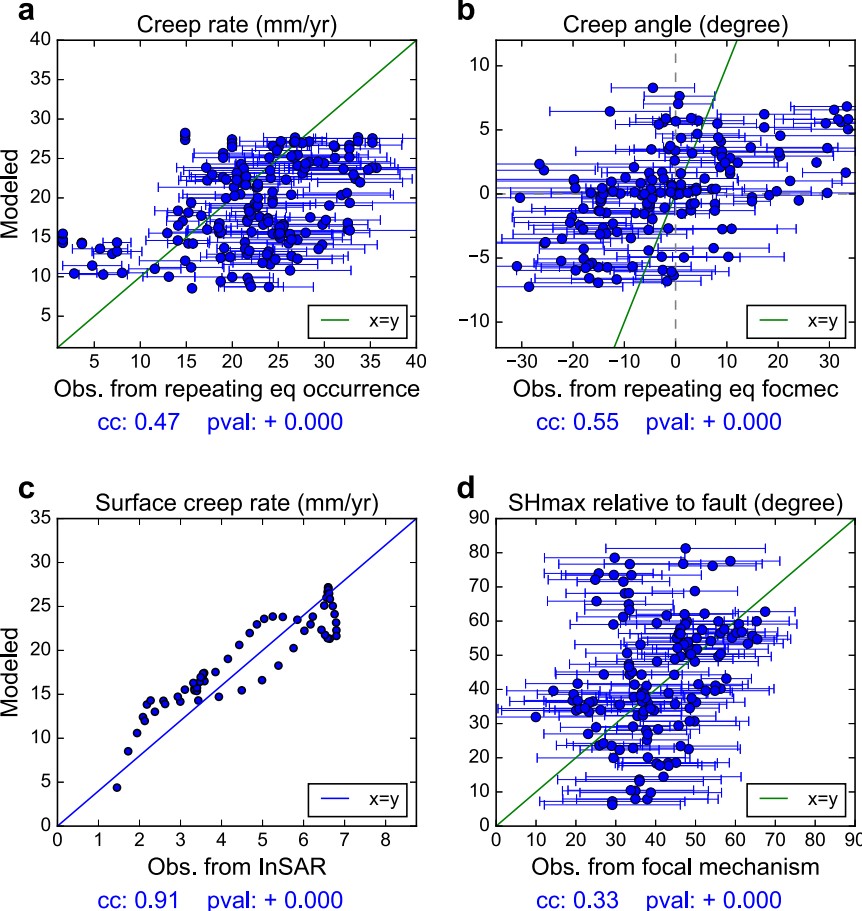

**Fig. 6 | Comparison between modeled and observed fault properties.** Point-to-point comparisons of the **a** creep rate, **b** creep direction, **c** surface creep rate, and **d** the angle between main fault and the off-fault maximum horizontal stress orientation ($\theta$) (from the area 1.5 km NE to the main fault) modeled from the fault coupling model A (Y-axes) and those estimated from **a** repeating earthquake occurrence rates, **b** repeating earthquake focal mechanisms, **c** InSAR LOS data, and **d** $M_l \geq 1.0$ focal mechanisms (X-axes). (See Figs. 4, 5b, c for the data used for comparison).

controversial question is whether major ruptures can dynamically penetrate deep into the central creeping section[5,23,41]. The deep locked patch near BW may weaken the barrier effect of the central creeping section and increase the possibility for major earthquakes to rupture through the whole creeping section. However, the stressing rate would be required to be high enough to penetrate multiple creeping sections between the small, locked patches along the CSAF. Note that while we can't fully rule out this scenario, there is no direct evidence that great ruptures have made it across the CSAF creeping section[41]. Therefore, a substantial portion of the accumulated moment deficit is more likely to be released by the afterslip of surrounding major earthquakes and moderate-magnitude earthquakes on the CSAF.

The fine-scale fault coupling heterogeneity can also affect the large-scale stress distribution by perturbing the stress field near the fault (Fig. 5a, b). The rotation of **SH$_{max}$** from high angles away from the main fault to low angles in the vicinity of the SAF has been observed and studied for decades[42–44]. Rice[45] proposed a high pore-pressure fracture zone model that can decrease the effective fault-normal compressive stress and cause stress rotation. However, this can only be applied to narrow, high pore-pressure fault zones[46,47]. Some studies explain the stress rotation as the combined effect of a weak fault and finite-width weak lower crust[46,48], which requires strong lateral variations of lower crustal properties. Scholz[47] suggested that the fault-parallel shear stress decreases with distance from the SAF due to the frictional resistance to strike-slip motion under a strong-fault hypothesis, which conflicts with the absence of

localized shear heating near the main fault[49,50]. In this study, our analysis revealed a remarkable divergence in stress distribution as modeled for the fully creeping model B (Supplementary Fig. 5) compared to those derived from model A and observed through focal mechanisms. This difference underscores that the substantial stress rotation observed across the fault is unlikely to be caused by large-scale aseismic creep. Instead, the stress rotations appear to be governed by fine-scale heterogeneity of the fault zone deformation that is closely associated with a mix of heterogeneous seismic/aseismic deformation processes, tectonic loading forces, and localized variation in rheological properties. Since the along-fault strike-slip motion and the fine-scale heterogeneous fault zone deformation generally co-exist along the whole SAF, this can also contribute to the observed stress rotation near the main fault and is not directly related to the strong- or weak-fault hypotheses.

In addition to stress orientation, the fine-scale fault zone coupling heterogeneity can also cause local stress concentrations around the major fault, facilitating multi-scale earthquake ruptures near the major fault and promoting the evolution of fault zone architecture. When earthquake source dimensions significantly exceed the fault zone width, their ruptures tend to align with the main fault orientation, showing consistent focal mechanisms ($M_l \geq 4$ earthquakes in Fig. 3e, f). When earthquake source dimensions are smaller than or comparable to the fault zone width, earthquakes in the fault zone exhibit more diverse focal mechanisms. However, they still tend to slip in a fault-parallel direction with a notable portion of secondary faults showing high angle to the main fault ($M_l$1-4

earthquakes in Figs. 3e, f, 5d, e). These small high-angle faults might be related to phenomena such as microfracturing near the fault tips, relay breaching and splay-faulting, promoting the weakening of rock adjacent to the developing fault[51]. The weakened rock in the fault zone potentially promotes the inclination of small subsidiary fault to slip along a fault-parallel direction, the linkage of fault segments, and the occurrences of moderate-and-large earthquakes along the major fault segments (Figs. 2 and 3)[51]. The narrow weak fault zone can also cause localized stress and slip concentrations, which may lead to a zone of localized stress rotation in the vicinity of the SAF. Since the narrow weak fault zone is much easier to deform than the surrounding strong host rock, moderate and large magnitude earthquakes along the CSAF are more likely to be right-lateral ruptures along the main fault orientation instead of complex ruptures that are frequently observed in wide, diffuse, and immature fault zones, like the 2019 $M_w$7.1 Ridgecrest earthquake[52] and the 2016 $M_w$7.8 Kaikoura earthquake[53].

One important thing to note is that neither the above-mentioned fault coupling distribution, the large-scale stress rotations, nor the relative weakness of the narrow fault zone indicate the absolute level of fault strength. Both frictional strength and fault coupling can be affected by variations of the stress field on the fault and variations of material properties, such as lithology[35–37], temperature[38,39], and pore fluid pressure[54]. However, there is no clear correlation between fault coupling and fault strength[55,56]. The variation of fault geometry and surface roughness[57] can also significantly affect the fault strength. One possible way to estimate the strength of the CSAF might be the direct estimation of absolute stress levels before moderate earthquake ruptures along the CSAF based on changes of small earthquake focal mechanisms[29]. For example, we can search over possible absolute stress levels, model the stress rotation in the fault zone before and after the 2004 Parkfield earthquake using finite source slip models[58,59], and compare the results with the focal mechanism observations[60].

In summary, we use high-resolution locations and focal mechanisms of repeating earthquakes, $M_l \geq 4$ earthquakes, and $M_l$1-4 earthquakes to illuminate fine-scale on-fault creep rates and directions as well as the stress field, structure, and earthquake slip variations in the fault zone. Our results reveal closely connected on-fault and off-fault deformation processes. All observed fine-scale kinematic features can be reconciled with a simple fault coupling model, inferred to be surrounded by a narrow, mechanically weak zone. Our study demonstrates the value of integrating small-earthquake focal mechanisms into fault zone analysis, to better resolve the detailed fault orientations, slip directions, and stress field variations associated with various aseismic and seismic processes. The resolved fault coupling heterogeneity, the surrounding narrow weak fault zone with complex internal structure, as well as their interactions have important implications for revealing multi-scale fault zone deformation, understanding large-scale stress field variation, and estimating the slip budget, timing, and patterns of future major earthquakes. The analyses performed in this study can be applied to other transform and subduction fault zones around the world for better understanding of multi-scale fault geometry and kinematics and improved estimation of future seismic hazard.

## Methods
### Fault slip modeling
The variation of fault displacement and stress around the fault depend on the mechanical response of the fault to the shear stress loading from the lower crust. Therefore, we used the boundary-element code Poly3D[31] to simulate fault slip processes during the inter-seismic period. The fault lies in a uniform elastic half-space with a Poisson's ratio of 0.25 and shear modulus of 30 GPa. It is vertical both in and below the seismogenic layer. We generate a triangular mesh for the fault surface for modeling and observe the results using $3 \times 3$ km grid for comparison with fault zone properties measured from data. The mesh comprises uniformly distributed triangles with each triangle having edge lengths of 1 km, 1.12 km and 1.12 km. In the top 15 km depth. Only zero-slip or zero-shear-stress boundary conditions on patches of the shallow fault are defined, rather than frictional properties. The fault is mostly freely slipping between −17 km and 166 km with several small frictionally locked patches and is fully coupled beyond 166 km NW and −17 km SE of the fault (model A; Fig. 4a). Between 15 km and 2000 km depth, the fault slips steadily with a 34 mm/yr inter-seismic slip rate between −500 km to 500 km horizontal distance along the fault. To evaluate the effect of the small, locked fault patched, we also consider a model without small frictionally locked patches between −17 km and 166 km along the fault but with a fully creeping fault bounded by locked segments at the two ends of the fault (model B; Supplementary Figs. 5, 6).

### $M_l \geq 1$ earthquake focal mechanism calculation
In this study, we calculate the focal mechanisms along the CSAF using the REFOC algorithm[29], which uses first-motion polarities, S-/P-wave amplitude ratios to obtain the initial earthquake focal mechanisms and further constrain them using the P-wave amplitude ratios and S-wave amplitude ratios of closely located earthquakes within 3 km hypocentral distance. We utilize the polarities, P- and S-wave phases manually picked by data analysts, and the relocated earthquake catalog archived by the Northern California Earthquake Data Center (NCEDC). We use the same P-wave velocity models[29] and the S-wave velocity models are derived from P-wave velocity models by assuming that the P-/S-wave velocity ratio equals 1.732. We apply a 1–10 Hz band-pass-filter to earthquake waveform data from NCEDC, choose 0.5 s before to 1.5 s after P- and S-wave arrivals as the signal windows and 2.5 s to 0.5 s before P-wave arrivals as noise windows, and take the difference between maximum and minimum amplitude values in each time window to be the estimated signal and noise amplitudes. If the time difference between P- and S-wave arrivals is larger than 2 s and the signal-to-noise ratio (SNR) is larger than 3, we use the P- and S-wave amplitudes to obtain S-/P-wave amplitude ratios for each individual event as well as the inter-event P-wave amplitude ratios and S-wave amplitude ratios. Given that $M_l \geq 3$ earthquakes have many recorded polarities and S-/P-wave amplitude ratios and considering that the corner frequencies of $M_l < 3$ earthquakes typically exceed 10 Hz, variations in corner frequency across earthquakes do not influence the estimation of inter-event P-wave amplitude ratios and S-wave amplitude ratios. Therefore, these observations can be used to improve the focal mechanism estimation. Finally, we obtained 52,211 out of 65,492 ( ~ 80%) $M_l \geq 1$ earthquakes with at least 8 polarities and focal mechanism uncertainties less than 35 degree. The catalog is available via (https://data.mendeley.com/datasets/34szj3jm6k/1).

### Repeating earthquake focal mechanism calculation
Since repeating earthquakes share highly similar waveforms and locations, they also share similar rupture processes and earthquake focal mechanisms. Therefore, we can utilize the similarity of repeating earthquakes to better constrain the focal mechanism of each repeating earthquake sequence. For each repeating earthquake sequence, we obtain available first-motion polarities and S-/P-wave amplitude ratios from all earthquakes in the sequence at each station, calculate the median values of polarities and S-/P-wave amplitude ratios, and assign the values to the station. Then we use the median polarities and S-/P-wave amplitude ratios to calculate the focal mechanism of each repeating earthquake sequence using the HASH algorithm[61]. By doing this process, we can both reduce the errors caused by manual picking and temporal noises in the waveforms and make full use of all available stations in the study time period for focal mechanism calculation. We

obtained focal mechanisms of 386 repeating earthquake sequences and the catalog is available via (https://data.mendeley.com/datasets/34szj3jm6k/1).

## Fault geometry estimation using repeating earthquakes and seismicity

In this study, we determine the main fault geometry (representative strike and dip and average fault-zone width of each segment) using earthquake locations at depth. Since small earthquakes may occur around the main fault instead of on-fault, we first determine the horizontal location of the primary fault strand using the locations of repeating earthquake sequences with at least two repeaters (red dots in Fig. 3a, b)[28], which generally indicate localized aseismic slip of major faults. We choose earthquakes within 1 km epicentral distance from the horizontal fault trace and calculate their median distance normal to XX' for each 3 km-long fault segment stepping at 1 km interval (Supplementary Fig. 16a). The obtained median value is used as the reference to calculate the relative distance normal to XX' for all chosen earthquakes. Supplementary Fig. 16b shows that most segments have events with either highly similar or gradual changes of distance normal to XX' across the whole depth range. Therefore, we assume that the fault can be treated as a 3D fault plane in each fault segment. We then used the chosen earthquakes to determine the 3D fault geometry by applying the principal component analysis[30] to minimize the orthogonal hypocentral distances to the fitted fault plane in each fault segment. For each 15 km-long and 15 km-deep fault segment stepping at 1 km intervals along the fault trace, we calculate the strike and dip of the plane that minimizes the distance between earthquake hypocenters and the plane and assign the values to the center of the fault segment (Fig. 3d).

## Fault creep rate estimation using repeating earthquakes

Repeating earthquakes are events that repeatedly rupture particular fault patches[16], which can be detected by waveform similarity[25,62] and can be used to illuminate the spatiotemporal variations of fault creep rate at depth[26,63]. Here, we use the repeating earthquake sequences with more than 10 repeaters from a Northern California repeating earthquake catalog[28] to estimate the creep rate variations along the CSAF.

The cumulative fault slip of and surrounding an earthquake patch over one seismic cycle in a repeating earthquake sequence can be estimated following the empirical scaling relationship

$$d = 10^{\alpha} M_0^{\beta}, \tag{1}$$

where $d$ is slip in centimeter and $M_0$ is seismic moment in dyne·cm, converted from the NCSN preferred magnitude $M_l$ using the empirical relationship[64]

$$\log(M_0) = 1.6 M_l + 15.8 \tag{2}$$

The empirical values and are $\alpha = -2.36$ and $\beta = 0.16$ based on comparison with the geodetically inferred creep rate at Parkfield[26]. To compare with the modeling results, we obtain the averaged slip rate of all repeating earthquakes in each $3 \times 3$ km fault patch in Fig. 4b and assign the value as the slip rate of the fault patch (Fig. 4c, Supplementary Figs. 17a, 18a). If the $3 \times 3$ km fault patch lacks repeating earthquake focal mechanisms, we will broaden our search to include a $9 \times 9$ km fault area centered around the same location. For each fault patch, we also employ a bootstrap method to estimate the uncertainties associated with the estimated fault creep rate based on 200 resampled focal mechanism datasets. Each resample is expected to contain approximately 80% unique data points, providing robust assessment of variability and confidence in the estimations.

## Local fault slip direction estimation using repeating earthquake focal mechanisms

Since repeating earthquakes are closely located around the main fault with strike orientation highly consistent with the main fault (Fig. 2), we assume that most repeating earthquakes are located on the main fault and represent the local fault slip behaviors. We first obtain the nodal plane whose strike angle has a smaller azimuthal difference from the main fault orientation. We then combine the dip and rake angles of the best-fitting nodal plane to estimate the slip direction of each repeating earthquake sequence (Supplementary Figs. 4, 16). Since we use the NE side of the main fault as the reference and the rake angle represent the moving direction of the hanging wall, we treat the rake direction as the slip direction when the nodal plane is dipping to the NE and use the opposite of the rake direction as slip direction when the nodal plane is dipping to the SW (Supplementary Fig. 17). To compare with the modeling results, we obtain the averaged slip direction of all repeating earthquakes in each $3 \times 3$ km fault patch in Fig. 4a and assign the value as the slip direction of the fault patch (Fig. 3e, Supplementary Fig. 18). If the $3 \times 3$ km fault patch lacks repeating earthquake focal mechanisms, we will broaden our search to include a $9 \times 9$ km fault area centered around the same location. For each fault patch, we also employ a bootstrap method to estimate the uncertainties associated with the estimated fault creep direction based on 200 resampled focal mechanism datasets. Each resample is expected to contain approximately 80% unique data points, providing robust assessment of variability and confidence in the estimations.

## Estimations of stress orientation (SH$_{max}$) and faulting style (%$_{rake>0}$) using $M_l \geq 1$ focal mechanisms

Focal mechanisms contain valuable information about fault geometry, kinematics and stress state in the crust. We use 24,915 $M_l \geq 1$ focal mechanisms located within 2 km from the main fault to estimate stress orientation SH$_{max}$ and fault style %$_{rake>0}$. We only use focal mechanisms with more than 8 polarities and <35 degrees uncertainties for quality control. To compare with the modeling results, we obtain these values in each $3 \times 3$ km fault patch in Fig. 4a when there are more than 100 focal mechanisms in the grid (Fig. 5b). If the $3 \times 3$ km fault patch lacks sufficient focal mechanisms, we will broaden our search to include a $9 \times 9$ km fault area centered around the same location.

For stress orientation estimation, we iteratively inverse stress using the STRESSINVERSE program[65]. This method is modified from Michael's method (1987) that jointly inverse stress and fault orientations by selecting the nodal plane with higher value of instability $I$[66]. For each fault patch, we introduce random perturbations to all input focal mechanisms according to their respective uncertainties. This process is repeated to calculate stress tensors 200 times for each set of perturbed focal mechanisms. The uncertainties of the resulting stress orientations are then quantified using the standard deviation of these 200 estimated stress tensors.

For earthquake faulting style, it is usually classified into normal, reverse, or strike-slip earthquakes based on rake angles[67]. Here, in order to represent the faulting style of a group of focal mechanisms, we simplify the classification into two types: oblique-reverse-faulting events with rake angle larger than 0° and oblique-normal-faulting events with rake angle equal to or smaller than 0° so that the summation of the percentages of oblique-normal-faulting and oblique-reverse-faulting events equals 100%. In this study, we use the percentage of oblique-reverse-faulting events (%$_{rake>0}$) to represent the earthquake faulting style.

## Reporting summary

Further information on research design is available in the Nature Portfolio Reporting Summary linked to this article.

## Data availability

The InSAR surface slip-rate estimates were obtained from Jolivet et al., refs. 11. The repeating earthquake catalog was obtained from Waldhauser and Schaff[28]. The relocated earthquake catalog was obtained from Waldhauser and Schaff[24]. The earthquake phase information and seismic waveforms are taken from the Northern California Earthquake Data Center, Northern California Seismic Network (https://ncedc.org/ncsn/). The estimated earthquake focal mechanisms are available at Mendeley Data (https://data.mendeley.com/datasets/34szj3jm6k/1).

## Code availability

The codes are available upon request to the authors.

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

## Acknowledgements

The authors thank Shiqing Xu, Taka'aki Taira, Naoki Uchida, and Saeko Kita for valuable comments and suggestions. This work was funded by the California Governor's Office of Emergency Services (Cal OES), Agreement Number 6172-2018. Y. C. acknowledges support from the Fundamental Research Funds for the Central Universities.

## Author contributions

Y.C. and R.B. designed the study. Y.C. processed all datasets and performed modeling and analyses. Y.C., R.B., and R.A. all contributed to discussions. Y.C. led the writing of the manuscript with contributions from R.B. and R.A.

## Competing interests

The authors declare no competing interests.
