## [Peer Review File · Nature Communications]

3D architecture and complex behavior along the simple central San Andreas faultREVIEWER COMMENTS

Reviewer #1 (Remarks to the Author):

In this paper, Cheng et al propose to explore the dense catalog of seismicity to derive new observables along the creeping section of the SAF. Although geodetic techniques allow to image slip at depth along such creeping sections, it is obvious that finer detail cannot be achieved from surface motion inversions. The authors propose then to finely observe the motion provided by earthquake focal mechanisms, considering independently on-fault repeaters and off-fault earthquakes to highlight fine crustal deformation underground.

The paper in itself is very innovative as I do not recall such observations along creeping faults. The large number of events recorded is essential and the ideas are really new. The newly developed catalog of focal mechanisms is impressive and will certainly be useful to many in future studies. Such approach could also be developed for other creeping faults globally, if seismic data allows it.

In general, the paper reads well, although I would suggest some slight reorganization to better emphasize the hypothesis tested in the paper. Figures nicely convey the message of the paper. However, I think that some additional justification is required to sustain the main message from the paper and illustrate the limits of the analysis. In particular, I believe some discrepancies between observations and models point to fundamental misconceptions and this should be better highlighted. I would suggest moderate revisions and I am certain this article will make a nice contribution to the study of aseismic slip along the SAF and elsewhere.

Main Comments:

- I strongly encourage the authors to re-write part of the paper in order to better emphasize the main hypothesis tested here: earthquake focal mechanisms provide crucial information on the finer details of fault locking in creeping sections. In particular, the model poly3D is described too late in the paper, which makes it difficult for readers to understand where do the simulations from figure 1 come from. Simply, these simulations suggest that if there is a locked patch at depth, we should observe a rotation in the rake of the aseismic slip vector as predicted by an elastic model and this cannot be resolved from a geodetic perspective. Since most people, me included, generally consider slip distributions as the mapping of geodetic data to slip on a fault, such rotations cannot be observed/trusted and this is the main tested hypothesis. I am not certain my point is clear enough, but some re-writing to better illustrate what we should expect and show that it is in the data would be useful. The same applies to the SHmax orientation.

- Although I am convinced there is a relationship between the modeled creep angle and the one inferred from repeaters, there is, as discussed in the discussion, a scaling between these two quantities. It would be nice to have some feeling about the local variability in observed creep angles within a single cell as well as the correlation itself. I hence think figure S6b should be moved to the main text. Furthermore, it is mentioned in the discussion that the mismatch between the range of modeled and observed angles is most likely due to local complexities, but I am not certain anything supports that claim. Would it be pertinent to illustrate this with a synthetic model relating the width/shape/aspect of a locked asperity

and slip rate to the amplitude of the predicted deviation? I would expect that for a more elongated and sharp asperity, creep angle changes would be larger, but this remains to be tested and could be added in the paper.

- The correlation between the modeled SHmax relative direction and the observed one is really not that obvious and I would moderate that point in the paper (0.33 is not “considerable”). What it might say is that aseismic slip along the creeping section does actually not have a significant influence on the state of stress off the main fault, which would indicate that such stress distribution is rather controlled by previous earthquakes, tectonic loading, local rheological heterogeneities or else, but not by aseismic slip. The correlation between the faulting style and the SHmax observed is better and allows some discussion though.

- It would be nice to have a visual aid to help readers to understand what a positive change in creep angle corresponds to (does the strike slip vector veer upwards or downwards?)

Minor comments:

- I am not certain the term “creep angle” is the best choice. It is a deviation from purely strike slip motion hence could it be called “creep deviation” or something else?

- We observe large variations in dip angle from the repeaters (Supp mat, 90° to 50°). Could you discuss this point? This would for instance suggest that normal stress on the fault might vary, which would in turn affect its weakness and potentially its ability to slide.

- Fig 1: You should add variations in “creep angle” in the vectors shown in Fig 1a to illustrate this encircling motion around locked asperities.

- Line 53, Page 3: It really depends from one model to the other.

- Line 55, Page 3: Why did you use this reference here (19)?

- Line 55, Page 3: “analysis of microseismicity...” I am not sure you mean “underlying physical properties of the fault zone” since this is not what the analysis of seismicity gives you. I would suggest “instead of the actual slip mechanism, which could allow eventually to infer some physical properties of the fault zone”, or something on that line.

- Line 59: Not sure why the reference to Fig 1 here.

- In this work...: I think your first hypothesis built from the simple modeling shown in Fig 1 should be better introduced here.

- Would there be a potential relationship between the creep angle variations and the encircling ruptures as described by Meng et al 2018 and Caballero et al 2023 for the Illapel earthquake? Is it the same thing?

- Line 158: The theta symbol should be used in the figures or not used in the text but some consistency between figures and text should be found.

- Line 171: The $\text{PercRake} > 0$ mathematical notation is not particularly easy to read.

- Line 207: You are saying too much and not enough at the same time. Would such type of inversion be feasible? Have you tried it? If written as is, I want to know more. Otherwise, maybe you should remove it.

- Line 241: Some foreshocks potentially coming from the creeping section are described before the Fort Tejon earthquake (see Sieh 1979). Could these asperities explain such earthquakes?

- Fig S12: I suspect some (all?) of the authors are also working on the Hayward fault, but Pt Pinole is not a usual reference point for the central SAF creeping section. Jokes aside, I am impressed by the nice first order consistency between these models.

I hope my comments will be of help to improve this interesting manuscript full of stimulating ideas,
Cheers,
Romain Jolivet, PhD
École normale supérieure

Reviewer #2 (Remarks to the Author):

This manuscript presents an interesting analysis of the seismic and aseismic deformation features along the central San Andreas Fault. The study supports the existence of complex fine-scale structure and kinematics of the fault zone that is mechanically weak and poorly coupled. I think the results can have a great impact on the estimation of future seismic hazard in the region, but I'm concerned about several parts of the analysis that may obscure what can be interpreted from the results.

My major comment is about the uncertainty in the focal mechanisms. In principle, the resolutions of nodal planes also depend on the magnitudes of earthquakes, with the nodal planes of smaller earthquakes harder to resolve. In other words, the larger variability of focal mechanisms of the $M > 1$ earthquakes may simply result from their larger uncertainties. It is mentioned in lines 103-104 that the percentages of $M > 4$ earthquakes and repeating earthquakes with less than 20 degree azimuthal difference are higher than $M > 1$ earthquakes, but this can be strongly affected by the large uncertainty in focal mechanisms of small earthquakes. Thus, I think the authors need to provide a thorough discussion about the uncertainties in nodal planes for earthquakes with different magnitudes (e.g., M1-2, M2-3, and M3-4). The focal mechanisms are also retrieved from data filtered at the same frequency band (1-10 Hz), which is lower than the corner frequencies of M1 or M2 earthquakes but include the corner frequencies of M3-4 earthquakes. How would this affect the resolution of focal mechanisms?

I'm also concerned about the estimation of fault geometry. The authors used 15-km-deep fault segments for the principal component analysis, which seems to imply that the fault geometry is the same along the 15 km depth, but the diverse focal mechanisms of $M > 4$ earthquakes suggest that the fault geometry vary with depths as well.

The model of fault displacements and the comparison between observed and modeled displacements are very interesting. Could the authors add more explanations about what affect the modeled creep rate and direction? They mentioned that the correlation coefficients are 0.47 and 0.55 between observed and modeled creep rate and direction (line 142), but I don't find the coefficients to be high enough to provide a strong support for the consistency of the models. Moreover, I think such correlation results also depend on the discretization of the model and it's hard to interpret the implications of such correlation coefficients.

I have similar concerns about the correlation coefficient between the modeled and observed stress field (line 161). The observed SHmax below SC and MM is also much smaller than the modeled values.

Lines 85-86: This description is confusing as the catalog of $M > 1$ earthquakes should include $M > 4$ earthquakes and some repeating earthquakes. Could you clarify this sentence?

Reviewer #1 (Remarks to the Author):

In this paper, Cheng et al propose to explore the dense catalog of seismicity to derive new observables along the creeping section of the SAF. Although geodetic techniques allow to image slip at depth along such creeping sections, it is obvious that finer detail cannot be achieved from surface motion inversions. The authors propose then to finely observe the motion provided by earthquake focal mechanisms, considering independently on-fault repeaters and off-fault earthquakes to highlight fine crustal deformation underground.

The paper in itself is very innovative as I do not recall such observations along creeping faults. The large number of events recorded is essential and the ideas are really new. The newly developed catalog of focal mechanisms is impressive and will certainly be useful to many in future studies. Such approach could also be developed for other creeping faults globally, if seismic data allows it.

In general, the paper reads well, although I would suggest some slight reorganization to better emphasize the hypothesis tested in the paper. Figures nicely convey the message of the paper. However, I think that some additional justification is required to sustain the main message from the paper and illustrate the limits of the analysis. In particular, I believe some discrepancies between observations and models point to fundamental misconceptions and this should be better highlighted. I would suggest moderate revisions and I am certain this article will make a nice contribution to the study of aseismic slip along the SAF and elsewhere.

We appreciate the positive and constructive assessment of our manuscript. In our revisions (see detailed responses below), we tried to further clarify our hypothesis and to better highlight some of the limits of the analysis and remaining discrepancies between data and models.

Main Comments:

- I strongly encourage the authors to re-write part of the paper in order to better emphasize the main hypothesis tested here: earthquake focal mechanisms provide crucial information on the finer details of fault locking in creeping sections. In particular, the model poly3D is described too late in the paper, which makes it difficult for readers to understand where do the simulations from figure 1 come from. Simply, these simulations suggest that if there is a locked patch at depth, we should observe a rotation in the rake of the aseismic slip vector as predicted by an elastic model and this cannot be resolved from a geodetic perspective. Since most people, me included, generally consider slip distributions as the mapping of geodetic data to slip on a fault, such rotations cannot be observed/trusted and this is the main tested hypothesis. I am not certain my point is clear enough, but some re-writing to better illustrate what we should expect and show that it is in the data would be useful. The same applies to the SHmax orientation.

Thank you! We have accordingly revised the Introduction to include a more detailed explanation of the simple elastic model depicted in Figure 1, as well as to emphasize the significance of employing focal mechanisms in our analysis.

(Line 59-76): “However, gaps remain in our understanding of the seismic and aseismic deformation of faults due to several problems. It is often assumed that all patches on a fault slip in the same direction without considering the variation of slip directions that we can expect in areas with abrupt changes of fault geometry or creep rate¹⁹. Secondly, analyses of seismicity often focus on statistical parameters estimated from earthquake occurrences instead of the underlying physical processes. Moreover, the effects of distributed fault-zone deformation and on-fault/off-fault interactions are usually ignored, although on-fault and off-fault deformations are tightly interlinked and coevolve with strong feedback loops over multiple spatial and temporal scales²⁰ (Fig. 1a). These challenges can be potentially overcome by analyzing focal mechanisms of small earthquakes, which helps to resolve fine-scale on-fault slip directions, provides physical parameters for seismicity analysis (e.g., fault and stress orientations), and allows for differentiating on- and off-fault seismic deformation (Fig. 1b). Furthermore, if a locked patch exists at depth, the elastic fault slip modeling results (see Methods section) show that the creep rate slightly decreases (Fig. 1c) and the creep direction rotates around the locked patch (Fig. 1d), suggesting that focal mechanisms can provide crucial information on the finer details of fault coupling that cannot be resolved from geodetically measured surface displacements. In this work, we integrate a comprehensive catalog of focal mechanisms to deepen our understanding of the fine-scale architecture and complex behaviors of major fault zones.”

- Although I am convinced there is a relationship between the modeled creep angle and the one inferred from repeaters, there is, as discussed in the discussion, a scaling between these two quantities. It would be nice to have some feeling about the local variability in observed creep angles within a single cell as well as the correlation itself. I hence think figure S6b should be moved to the main text.

(Line 807-813) Thanks! We have moved Figure S6b into the main text as the new Figure 6 and now discuss this figure in detail in the second paragraph of the Discussion section.

(Line 169-172) “The modeled results are overall consistent with the observed subsurface fault creep rate variations, the fault slip directions at depth, and the surface creep rates with correlation coefficients of 0.47 (Fig. 4b-c), 0.55 (Fig. 4d-e), and 0.91 (Fig. 4f), respectively (Fig. 6a, 6b, 6c).”

(Line 188-192) “We also observe a moderate correlation (0.33) between the modeled off-fault stress distribution and the observed stress field estimated using $M_l \geq 1.0$ focal mechanisms located within 2 km of the main fault (Figs. 5b and 6d), which is noteworthy considering the additional effects of background tectonic stress and the strong stress variations across the fault.”

(Line 242-244) “The range of modeled creep directions is ± 6 degrees from the horizontal direction and is substantially smaller than that of the creep directions observed from repeating earthquake focal mechanisms (± 30 degrees) (Fig. 6b).”

(Line 253-255) “In contrast to the fault creep direction, the range of the modeled angle between SH_{\max} and fault strike direction (θ) is 5-85 degrees is larger than the observed range (20-70 degrees) (Fig. 6d).”

Furthermore, it is mentioned in the discussion that the mismatch between the range of modeled and observed angles is most likely due to local complexities, but I am not certain anything supports that claim. Would it be pertinent to illustrate this with a synthetic model relating the width/shape/aspect of a locked asperity and slip rate to the amplitude of the predicted deviation? I would expect that for a more elongated and sharp asperity, creep angle changes would be larger, but this remains to be tested and could be added in the paper.

We have carried out several synthetic model tests described as follows in Line 242-253.

“The range of modeled creep directions is ± 6 degrees from the horizontal direction and is substantially smaller than that of the creep directions observed from repeating earthquake focal mechanisms (± 30 degrees) (Fig. 6b). To explore the possible reasons for the observed misfits, we have performed a series of additional synthetic tests. The results show that locked patches can cause stronger creep direction rotation when the patch is shallower (Supplementary Fig. 7) or larger (Supplementary Figs. 8 and 9), and the adjacent freely creeping area is larger (Supplementary Fig. 10). Many other factors that can intensify creep direction rotation are not captured in our model, such as deviations of fault segments from the vertical orientation (Supplementary Fig. 11), and the more substantial rotations near the locked patch’s edge using models with finer grid sizes (Supplementary Fig. 12). The inferred rake angle on the fault plane can also change substantially due to local geometry variations and focal mechanism uncertainties (Hardebeck and Shearer, 2002; ~ 20 degree in this study), which can further affect the inferred creep direction.”

- The correlation between the modeled SH_{\max} relative direction and the observed one is really not that obvious and I would moderate that point in the paper (0.33 is not “considerable”).

Thanks! We have updated the description in the “Off-fault structure and kinematics” part:

(Line 188-192) “We also observe a moderate correlation (0.33) between the modeled off-fault stress distribution and the observed stress field estimated using $M_l \geq 1.0$ focal mechanisms located within 2 km of the main fault (Figs. 5b and 6d), which is noteworthy considering the additional effects of background tectonic stress and the strong stress variations across the fault.”

What it might say is that aseismic slip along the creeping section does actually not have a significant influence on the state of stress off the main fault, which would indicate that such stress distribution is rather controlled by previous earthquakes, tectonic loading, local rheological heterogeneities or else, but not by aseismic slip. The correlation between the faulting style and the SH_{\max} observed is better and allows some discussion though.

Sure, that is a good suggestion! We have updated the Discussion section accordingly. (Line 345-358) “In this study, our analysis revealed a remarkable divergence in stress distribution as modeled for the fully creeping model B (Supplementary Fig. 5) compared to those derived from model A and observed through focal mechanisms. This difference underscores that the substantial stress rotation observed across the fault is unlikely to be caused by large-scale aseismic creep. Instead, the stress rotations appear to be governed by fine-scale heterogeneity of the fault zone deformation that is closely associated with a mix of heterogeneous seismic/aseismic deformation processes, tectonic loading forces, and localized variation in rheological properties. Since the along-fault strike-slip motion and the fine-scale heterogeneous fault zone deformation generally co-exist along the whole SAF, this can also contribute to the observed stress rotation near the main fault and is not directly related to the strong- or weak-fault hypotheses.”

- It would be nice to have a visual aid to help readers to understand what a positive change in creep angle corresponds to (does the strike slip vector veer upwards or downwards?)

Thanks! We have updated the Figs 1d and 4d highlighting the slip vector by adding scale legend labels and black lines denoting the local creep direction on the NE side of the fault.

Minor comments:

- I am not certain the term “creep angle” is the best choice. It is a deviation from purely strike slip motion hence could it be called “creep deviation” or something else?

Thank you for your valuable suggestion! We have revised the terminology to 'creep direction' and enhanced the clarity by adding detailed explanations such as '* degrees up/down' or '0 degree horizontal' next to the color bar ticks. (Fig. 1, Fig 4)

- We observe large variations in dip angle from the repeaters (Supp mat, 90° to 50°). Could you discuss this point? This would for instance suggest that normal stress on the fault might vary, which would in turn affect its weakness and potentially its ability to slide.

Thanks for the suggestion! We have added more discussion about the possible influence of the first-order observation of dip angle variation inferred from repeating earthquakes.

(Line 306-309) “In contrast, the local dip angle estimated from repeating earthquakes near SJB and PK exhibit about 20-30 degrees deviation from the major fault orientation, suggest that local fault geometry variation might considerably affect the fault coupling near the SJB and PK (Supplementary Fig. 15).”

- Fig 1: You should add variations in “creep angle” in the vectors shown in Fig 1a to illustrate this encircling motion around locked asperities.

Thanks! We have updated the Figs 1d and 4d to highlight the slip vector.

- Line 53, Page 3: It really depends from one model to the other.

(Line 60) Yes, thank you for your insightful comments! We agree that the applicability of this statement can vary across different models. This variability is precisely why we have used the term 'often' in our description at Line 53, Page 3, to acknowledge that while this observation is generally true, there are exceptions depending on the specific model in question.

- Line 55, Page 3: Why did you use this reference here (19)?

(Line 62) Because in this paper, Bürgmann et al., found that the observed slip rate anomaly from InSAR measurement can be explained by a vertical slip component of the fault and they also modeled this using the boundary element method, which supports the argument of the varying slip direction on the fault.

- Line 55 (Line 53), Page 3: “analysis of microseismicity...” I am not sure you mean “underlying physical properties of the fault zone” since this is not what the analysis of seismicity gives you. I would suggest “instead of the actual slip mechanism, which could allow eventually to infer some physical properties of the fault zone”, or something on that line.

Thanks for your suggestion! We have added more explanations about fault zone physical parameters and architecture below.

(Line 66-70) “These challenges can be potentially overcome by analyzing focal mechanisms of small earthquakes, which helps to resolve fine-scale on-fault slip directions, provides physical parameters for seismicity analysis (e.g., fault and stress orientations), and allows for differentiating on- and off-fault seismic deformation (Fig. 1b).”

Seismicity analysis does not directly reveal underlying physical processes; rather, it offers observations of slip behavior from which the slip mechanism can be inferred. Therefore, we have revised this section accordingly:

(Line 62-63) “Secondly, analyses of seismicity often focus on statistical parameters estimated from earthquake occurrences instead of the underlying physical processes.”

- Line 59: Not sure why the reference to Fig 1 here.

(Line 66) Because Figure 1a shows geodetic observations, earthquake locations, and the horizontal slip direction assumption that are used in traditional analysis. We added figure 1b for further comparison.

(Line 66-70) “These challenges can be potentially overcome by analyzing focal mechanisms of small earthquakes, which helps to resolve fine-scale on-fault slip directions, provides physical

parameters for seismicity analysis (e.g., fault and stress orientations), and allows for differentiating on- and off-fault seismic deformation (Fig. 1b).”

- In this work...: I think your first hypothesis built from the simple modeling shown in Fig 1 should be better introduced here.

Sure! Thanks for your suggestion! We have added more explanation about this model here.

(Line 70-76) “Furthermore, if a locked patch exists at depth, the elastic fault slip modeling results (see Methods section) show that the creep rate slightly decreases (Fig. 1c) and the creep direction rotates around the locked patch (Fig. 1d), suggesting that focal mechanisms can provide crucial information on the finer details of fault coupling that cannot be resolved from geodetically measured surface displacements. In this work, we integrate a comprehensive catalog of focal mechanisms to deepen our understanding of the fine-scale architecture and complex behaviors of major fault zones.”

- Would there be a potential relationship between the creep angle variations and the encircling ruptures as described by Meng et al 2018 and Caballero et al 2023 for the Illapel earthquake? Is it the same thing?

While there may be a connection, it is important to clarify that these phenomena are not identical. The models in these studies are designed for co-seismic slip, and despite being kinematic in nature, they are significantly influenced by the dynamics of propagating major ruptures, unlike our interseismic fault slip study.

- Line 158 (line 187): The theta symbol should be used in the figures or not used in the text but some consistency between figures and text should be found.

Thanks! We have updated the text and now the symbol and its meaning are mentioned in the main text (Line 192-203; Line 207-217)! We have also updated the labels in Fig 5 accordingly.

- Line 171 (line 208): The $\text{PercRake}>0$ mathematical notation is not particularly easy to read.

Thanks for your suggestion! We have updated it to $\%_{rake>0}$ both in the text and figures.

- Line 207: You are saying too much and not enough at the same time. Would such type of inversion be feasible? Have you tried it? If written as is, I want to know more. Otherwise, maybe you should remove it.

Thanks for the suggestion! We have removed this statement.

- Line 241: Some foreshocks potentially coming from the creeping section are described before the Fort Tejon earthquake (see Sieh 1979). Could these asperities explain such earthquakes?

Yes. Supplementary Fig. 1 shows historical $M > 5.5$ earthquakes including foreshocks of the 1857 mainshock.

We have updated the text accordingly.

(Line 318-323) “Since the locked patches are distributed along the creeping section with low slip deficit around them, they are more likely to rupture independently as moderate earthquakes (e.g., the foreshocks of the 1857 Fort Tejon earthquake and repeated occurrences of M6 Parkfield earthquakes; Supplementary Fig. 1) with a substantial amount of slip deficit on the surrounding fault being taken up by transient afterslip of major ruptures (e.g., following the 1906 Great San Francisco, 1989 Loma Prieta, and 2004 Parkfield earthquakes^{15,27}).”

- Fig S12 (Fig S14): I suspect some (all?) of the authors are also working on the Hayward fault, but Pt Pinole is not a usual reference point for the central SAF creeping section. Jokes aside, I am impressed by the nice first order consistency between these models.

Sorry for the “funny” typo and thanks for catching it! We have updated the label of the figure 14 in the supplementary material.

I hope my comments will be of help to improve this interesting manuscript full of stimulating ideas,

Cheers,

Romain Jolivet, PhD

École normale supérieure

Reviewer #2 (Remarks to the Author):

This manuscript presents an interesting analysis of the seismic and aseismic deformation features along the central San Andreas Fault. The study supports the existence of complex fine-scale structure and kinematics of the fault zone that is mechanically weak and poorly coupled. I think the results can have a great impact on the estimation of future seismic hazard in the region, but I'm concerned about several parts of the analysis that may obscure what can be interpreted from the results.

My major comment is about the uncertainty in the focal mechanisms. In principle, the resolutions of nodal planes also depend on the magnitudes of earthquakes, with the nodal planes of smaller earthquakes harder to resolve. In other words, the larger variability of focal mechanisms of the $M > 1$ earthquakes may simply result from their larger uncertainties. It is mentioned in lines 103-104 that the percentages of $M > 4$ earthquakes and repeating earthquakes with less than 20 degree azimuthal difference are higher than $M > 1$ earthquakes, but this can be strongly affected by the large uncertainty in focal mechanisms of small earthquakes. Thus, I think the authors need to provide a thorough discussion about the uncertainties in nodal planes for earthquakes with different magnitudes (e.g., M1-2, M2-3, and M3-4).

Thank you for your suggestion. In response, we have included a new figure in the supplementary material that illustrates the relationship between focal mechanism

uncertainty and earthquake magnitude. It is evident from this addition that focal mechanism uncertainty does indeed increase as the magnitude decreases. However, it is important to note that our study exclusively utilizes earthquake data where the focal mechanism uncertainty is below 25 degrees. Furthermore, the observed decrease in diversity in focal mechanisms with increasing magnitudes between M3.0 to M5.0 suggests that this phenomenon is attributable to genuine physical processes rather than being merely a byproduct of data uncertainties.

(Line 120-125) “To better quantify the fault zone structure, we **select all focal mechanism solutions with uncertainty less than 25 degrees** and pick the focal mechanism nodal plane closer to the main fault orientation and calculate the azimuthal difference between the nodal plane and the main fault.”

(Line 130-132) “**The diversity of earthquake focal mechanisms increases gradually with decreasing magnitude even between M_l 3.0 to 5.0 (Supplementary Fig. 3), suggesting that this trend is not caused by the uncertainty of focal mechanism solutions.**”

The focal mechanisms are also retrieved from data filtered at the same frequency band (1-10 Hz), which is lower than the corner frequencies of M1 or M2 earthquakes but include the corner frequencies of M3-4 earthquakes. How would this affect the resolution of focal mechanisms?

Thank you for your insightful comment! The determination of the focal mechanism primarily relies on the analysis of first-motion polarity, with additional constraints provided by the S/P amplitude ratios and the inter-event P/P and S/S amplitude ratios.

(Line 444-448) “**Given that $M_l > 3$ earthquakes have many recorded polarities and S-/P-wave amplitude ratios and considering that the corner frequencies of $M_l < 3$ earthquakes typically exceed 10Hz, variations in corner frequency across earthquakes do not influence the estimation of inter-event P-wave amplitude ratios and S-wave amplitude ratios. Therefore, these observations can be used to improve the focal mechanism estimation.**”

I'm also concerned about the estimation of fault geometry. The authors used 15-km-deep fault segments for the principal component analysis, which seems to imply that the fault geometry is the same along the 15 km depth, but the diverse focal mechanisms of $M > 4$ earthquakes suggest that the fault geometry vary with depths as well.

Thank you for your feedback. As illustrated in Figures 2a, 3a, and 3f, the focal mechanisms of earthquakes with magnitudes greater than 4 exhibit remarkable consistency, with over 90% showing very small azimuthal differences from the main fault. This observation suggests a high degree of planarity in the major fault geometry. However, due to the scarcity of earthquakes with magnitudes greater than 4, determining variations in fault geometry at different depths remains challenging.

To further explore this aspect, we analyzed the relative changes in distance normal to the line XX' at various depths (see Supplementary Fig. 16), finding that the fault geometry is generally near vertical or exhibits very gradual changes across different depths. This finding is corroborated by cross-sectional views of each fault segment

presented in Supplementary Fig. 2. For additional details, please refer to the Methods section.

(Line 466-486)

“Fault geometry estimation using repeating earthquakes and seismicity

In this study, we determine the main fault geometry (representative strike and dip and average fault-zone width of each segment) using earthquake locations at depth. Since small earthquakes may occur around the main fault instead of on-fault, we first determine the horizontal location of the primary fault strand using the locations of repeating earthquake sequences with at least two repeaters (red dots in Fig. 3a-b)²⁸, which generally indicate localized aseismic slip of major faults. We choose earthquakes within 1-km epicentral distance from the horizontal fault trace and calculate their median distance normal to XX' for each 3-km-long fault segment stepping at 1-km interval (Supplementary Fig. 16a). The obtained median value is used as the reference to calculate the relative distance normal to XX' for all chosen earthquakes. Supplementary Fig. 16b shows that most segments have events with either highly similar or gradual changes of distance normal to XX' across the whole depth range. Therefore, we assume that the fault can be treated as a 3D fault plane in each fault segment. We then used the chosen earthquakes to determine the 3D fault geometry by applying the principal component analysis³⁰ to minimize the orthogonal hypocentral distances to the fitted fault plane in each fault segment. For each 15-km-long and 15-km-deep fault segment stepping at 1-km intervals along the fault trace, we calculate the strike and dip of the plane that minimizes the distance between earthquake hypocenters and the plane and assign the values to the center of the fault segment (Fig. 3d).”

The model of fault displacements and the comparison between observed and modeled displacements are very interesting. Could the authors add more explanations about what affect the modeled creep rate and direction? They mentioned that the correlation coefficients are 0.47 and 0.55 between observed and modeled creep rate and direction (line 142), but I don't find the coefficients to be high enough to provide a strong support for the consistency of the models. Moreover, I think such correlation results also depend on the discretization of the model and it's hard to interpret the implications of such correlation coefficients.

Thank you for your comment! In this study, we have employed multiple observational datasets and conducted simple forward modeling, rather than engaging in complex inversions to optimize the model parameters. Given this approach, achieving a perfect correlation between all modeled outcomes and observations from various perspectives was not the objective and would be inherently challenging. This complexity is particularly pronounced when considering that creep rate and creep direction constitute two-dimensional observations, as opposed to the one-dimensional surface creep rate observations. Additionally, the fault creep direction and rate are influenced by local fault complexities, which our model does not fully capture. Consequently, we believe that correlation coefficients of 0.47 and 0.55 represent substantial correlations for a model that tries to account for the 3D distribution of multiple observations across a large fault system spanning 190 km.

(Line 244-253) “To explore the possible reasons for the observed misfits, we have performed a series of additional synthetic tests. The results show that locked patches can cause stronger creep direction rotation when the patch is shallower (Supplementary Fig. 7) or larger (Supplementary Figs. 8 and 9), and the adjacent freely creeping area is larger (Supplementary Fig. 10). Many other factors that can intensify creep direction rotation are not captured in our model, such as deviations of fault segments from the vertical orientation (Supplementary Fig. 11), and the more substantial rotations near the locked patch’s edge using models with finer grid sizes (Supplementary Fig. 12). The inferred rake angle on the fault plane can also change substantially due to local geometry variations and focal mechanism uncertainties (Hardebeck and Shearer, 2002; ~20 degree in this study), which can further affect the inferred creep direction.”

I have similar concerns about the correlation coefficient between the modeled and observed stress field (line 161) (Line 188). The observed SHmax below SC and MM is also much smaller than the modeled values.

Thank you for your insightful comment! The stress field, reflecting long-term deformation, is notably more complex than other parameters due to its high sensitivity to local rheological variations. It is challenging to attribute the inferred stress distribution solely to the interseismic deformation for the purpose of inverting the fault coupling model.

While the misfit below SC and MM is relatively pronounced, Model A exhibits a significantly smaller misfit compared to Model B. This observation underscores the critical influence of local fault coupling heterogeneity on stress rotation. In light of the moderate yet statistically significant correlation between the modeled and observed stress fields, we have revised our terminology regarding the correlation coefficient (CC) to "noteworthy." Additionally, we have updated our discussion of the stress rotation to acknowledge that while fault coupling heterogeneity contributes to stress rotation, it may not be the predominant factor.

(Line 188-192) “We also observe a moderate correlation (0.33) between the modeled off-fault stress distribution and the observed stress field estimated using $M_l \geq 1.0$ focal mechanisms located within 2 km of the main fault (Figs. 5b and 6d), which is noteworthy considering the additional effects of background tectonic stress and the strong stress variations across the fault.”

(Line 345-358) “In this study, our analysis revealed a remarkable divergence in stress distribution as modeled for the fully creeping model B (Supplementary Fig. 5) compared to those derived from model A and observed through focal mechanisms. This difference underscores that the substantial stress rotation observed across the fault is unlikely to be caused by large-scale aseismic creep. Instead, the stress rotations appear to be governed by fine-scale heterogeneity of the fault zone deformation that is closely associated with a mix of heterogeneous seismic/aseismic deformation processes, tectonic loading forces, and localized variation in rheological properties. Since the along-fault strike-slip motion and the fine-scale heterogeneous fault zone deformation generally co-exist along the whole SAF, this can also contribute to the

observed stress rotation near the main fault and is not directly related to the strong- or weak-fault hypotheses.”

Lines 85-86: This description is confusing as the catalog of $M > 1$ earthquakes should include $M > 4$ earthquakes and some repeating earthquakes. Could you clarify this sentence?

Thanks! We have added the following explanation.

(Line 127-130) “For these near-fault earthquakes, we obtain the percentages of $M_l \geq 4.0$ earthquakes, repeating earthquake sequences, and all $M_l \geq 1.0$ earthquakes (including $M_l \geq 4.0$ earthquakes and repeating earthquakes) with less than 20° azimuthal difference from the main fault, which are 92.9%, 94.1%, and 83.3%, respectively (Fig. 3f).”

REVIEWERS' COMMENTS

Reviewer #1 (Remarks to the Author):

I have now read and assessed the rebuttal letter and the new manuscript provided by the authors. I believe the paper is now in a much better shape and gives justice to the really innovative results shown here. To me, it could be accepted almost as is. I have two very minor points that do not require further examination from my side.

Page 11: You refer to a locked patch at depth which is visible in all geodetic models and I have the impression there is some mixup between high and low coupling somewhere. Please double check consistency.

Figure 6: The comparison between the InSAR data and the modeled surface creep shows very different values. It should be written somewhere that modeled horizontal motion is converted to LOS motion.

Looking forward to see this paper published,

Cheers,

Romain Jolivet

École normale supérieure

Reviewer #2 (Remarks to the Author):

I appreciate all the detailed responses to my previous comments and the clarifications incorporated in the manuscript. I think my concerns have been adequately addressed and the revised manuscript is suitable for publication.

REVIEWERS' COMMENTS

Reviewer #1 (Remarks to the Author):

I have now read and assessed the rebuttal letter and the new manuscript provided by the authors. I believe the paper is now in a much better shape and gives justice to the really innovative results shown here. To me, it could be accepted almost as is. I have two very minor points that do not require further examination from my side.

Page 11: You refer to a locked patch at depth which is visible in all geodetic models and I have the impression there is some mixup between high and low coupling somewhere. Please double check consistency.

Thanks for your careful checking! We have updated it to “high-coupling area” at line 231.

Figure 6: The comparison between the InSAR data and the modeled surface creep shows very different values. It should be written somewhere that modeled horizontal motion is converted to LOS motion.

Thanks! We have added InSAR LOS data in the caption of fig. 4 and fig.6

Looking forward to see this paper published,
Cheers,
Romain Jolivet
École normale supérieure

Reviewer #2 (Remarks to the Author):

I appreciate all the detailed responses to my previous comments and the clarifications incorporated in the manuscript. I think my concerns have been adequately addressed and the revised manuscript is suitable for publication.

Thanks for your positive feedback!